# Benefits and Pitfalls of a Glycosylation Inhibitor Tunicamycin in the Therapeutic Implication of Cancers

**DOI:** 10.3390/cells13050395

**Published:** 2024-02-25

**Authors:** Snigdha Banerjee, Affan A. Ansari, Sunil P. Upadhyay, Daniel J. Mettman, Jamie R. Hibdon, Mohiuddin Quadir, Pratyusha Ghosh, Anjali Kambhampati, Sushanta K. Banerjee

**Affiliations:** 1Cancer Research Unit, VA Medical Center, Kansas City, MO 64128, USA; aaansari@bluevalleyk12.net (A.A.A.); sunil.upadhyay@va.gov (S.P.U.); daniel.mettman@va.gov (D.J.M.); jamie.hibdon@va.gov (J.R.H.); anjalik07@icloud.com (A.K.); 2Department of Pathology and Laboratory Medicine, University of Kansas Medical Center, Kansas City, KS 66160, USA; 3Pathology Department, City VA Medical Center, Kansas City, MO 64128, USA; 4Department of Coatings and Polymeric Materials, North Dakota State University, Fargo, ND 58108, USA; mohiuddin.quadir@ndsu.edu (M.Q.); pratyusha.ghosh@ndsu.edu (P.G.)

**Keywords:** tunicamycin, breast cancer, nanoparticle, immunotherapy, glycosylation, drug resistance, multidrug therapy

## Abstract

The aberrant glycosylation is a hallmark of cancer progression and chemoresistance. It is also an immune therapeutic target for various cancers. Tunicamycin (TM) is one of the potent nucleoside antibiotics and an inhibitor of aberrant glycosylation in various cancer cells, including breast cancer, gastric cancer, and pancreatic cancer, parallel with the inhibition of cancer cell growth and progression of tumors. Like chemotherapies such as doxorubicin (DOX), 5′fluorouracil, etoposide, and cisplatin, TM induces the unfolded protein response (UPR) by blocking aberrant glycosylation. Consequently, stress is induced in the endoplasmic reticulum (ER) that promotes apoptosis. TM can thus be considered a potent antitumor drug in various cancers and may promote chemosensitivity. However, its lack of cell-type-specific cytotoxicity impedes its anticancer efficacy. In this review, we focus on recent advances in our understanding of the benefits and pitfalls of TM therapies in various cancers, including breast, colon, and pancreatic cancers, and discuss the mechanisms identified by which TM functions. Finally, we discuss the potential use of nano-based drug delivery systems to overcome non-specific toxicity and enhance the therapeutic efficacy of TM as a targeted therapy.

## 1. Introduction

Protein glycosylation is a unique physiological and pathophysiological process that attaches glycans (carbohydrate-based polymers) to the protein backbone via N-linked glycosylation (at the asparagines) or O-linked glycosylation (at the hydroxyl group of serine/threonine side chains) [1,2,3,4]. Glycosylation is the most frequent post-translational modification of a protein [5] and the key molecular event associated with innate and adaptive immune systems [6]. It is involved in cell membrane formation, folding and unfolding, and protein stability [3]. Thus, the functional utilities of protein glycosylation in health and diseases are of growing interest in biomedical research [6,7].

The aberrant glycosylation with unique structures, known as tumor-associated carbohydrate antigens (TACAs) [8,9,10], plays a vital role in malignant transformation, cancer progression, and metastasis [11,12]. Aberrant glycosylation, mainly associated with enzymes (glycosyltransferases and glycosidase), is useful as a biomarker for cancer cells. The enzymes that cause aberrant glycosylation express cancer-specific changes, especially the overexpression of certain enzymes in cellular biosynthesis. For example, in the study of proteomics, analysis of the glycans and glycoproteins in cancer and normal cells can yield personalized diagnosis and treatments, partly because the metabolism of cancer cells differs from that of normal cells. By continuing to study the cancer-modified glycans, these biomarkers can be better used to detect and diagnose specific types of cancer [13]. The spreading of primary cancer cells to distant organs is coupled with the alterations and degradation of the extracellular matrix (ECM), impacting cellular interactions and structural changes of surface proteins by glycosylation [3,6,7,14,15,16,17]. Thus, glycosylation is now reconsidered in the era of cancer-targeted therapy [18]. Thereby, particular emphasis on glycosylation inhibitors is given for the rational development of immunologically targeted cancer therapies.

TM, a mixture of homologous nucleoside antibiotic inhibitors of N-linked glycosylation, may inhibit tumor growth and resistance in multidrug therapy. This drug impedes the transfer of UDP-N-acetylglucosamine (GlcNAc) to dolichol phosphate in the endoplasmic reticulum (ER) of eukaryotic cells, thus disrupting protein maturation by blocking oligosaccharides biosynthesis [19]. Oligosaccharides are instrumental in N-linked glycosylation, linking to nitrogen atoms from asparagine residue. The formulation and structure of proteins depend on this natural process. The inhibition of this process results in unfolded protein response (UPR) and cellular stress, resulting in cell death/apoptosis via multiple signaling pathways. These include possibly activating the mTORC1-eNOS-RagC pathway [20] and TRAIL-induced pathway [21]. However, as effective as it is, this stalwart offensive of canceling N-linked glycosylation is not cell-type-specific. TM in cancer treatment could prove hazardous, as residual cytotoxicity on surrounding normal tissue or organs could worsen the state of the disease. Nanocarriers, a prospective asset in the pharmaceutical world, may present a solution to hinder this drug’s cytotoxicity.

## 2. Origin and Basic Biology of Tunicamycin

TM is a nucleoside antibiotic consisting of tunicamine sugar, uracil, N-acetylglucosamine, and the fatty acyl side chain (Figure 1). *Streptomyces lysosuperificus*, classified by Tamura et al. in 1970, is the sole producer of this antibiotic [22]. TM was detected as a product of fermentation of *Streptomyces lysosuperificus* and accumulated over 100 h. Newcastle disease virus was used as a subject to test the antibiotic activity. TM exhibits antibacterial activity due to its inhibition of phospho-MurNAc-pentapeptide translocase (MraY), an enzyme essential in bacterial cell wall synthesis [23]. MraY is classified as a translocase that helps in cell wall biosynthesis utilizing UDP-MurNAc, a pentapeptide necessary in peptidoglycan synthesis. TM inhibits MraY similarly to how it blocks dolichol phosphate in N-linked glycosylation. In a study conducted by Duskin et al., three variants of 3T3 fibroblasts were treated with TM: normal 3T3 cells, SV40-transformed 3T3 cells, and polyomavirus-transformed 3T3 cells [24]. TM was administered to variants in cell plates and displayed high cytotoxic effects on all 3T3 cell types, differing in some factors. After 2–4 days, the cytotoxicity of TM altered the morphology of their regular epithelioid shapes into long, pointed structures. These changes were observed when TM was present in all cell plates. However, when TM was removed, any physiological changes soon regressed into a normal state, proving that TM lingers around any targeted subject and applies cytotoxicity until removed from a subject.

## 3. How Does Tunicamycin Work?

A multifaceted mode of action of TM has been reported as a therapeutic drug. In any cancer treatment, the development of drug resistance seems to be almost imminent, and this limiting factor prevents the achievement of cures in patients with cancers [25]. The aberrant glycosylation is one of the leading factors in developing drug resistance phenotypes in cancer cells [26]. Some oncoproteins and oncogenes promote aggressive phenotypes such as proliferation and drug resistance via N-linked glycosylation [27,28]. TM inhibits N-linked glycosylation and other N-glycans via inducing ER stress and hindering the development of invasive behavior of cancer cells. These include cancer stemness and drug resistance [29]. However, it is uncertain if TM can regulate the mutant effect of proto-oncogenes associated with the above pathological events. Mechanistically, TM can enter eukaryotes via the protein transporter major facilitator domain containing 2A (MFSD2A) and permeates through the endoplasmic reticulum (ER) membrane, causing stress [30,31]. TM induces ER stress through the UPR, promoting TRAIL-induced apoptosis [20,32]. The whole genome microarray analysis found that, under sustained ER stress by TM, endothelial nitric oxide synthase (eNOS) upregulated in prostate cancer cells that activated mTORC1 through the eNOS-RagC pathway and promote p62-reactive oxygen species (ROS)-dependent mitochondrial apoptosis [20]. The UPR work through three receptors are located on the ER membrane. These include protein kinase RNA-like endoplasmic reticulum kinase (PERK), inositol-requiring enzyme (IRE1), and activating transcription factor 6 (ATF6) [33,34,35,36,37]. These receptors are usually found inactive in scenarios where UPR is nonexistent due to its regulation by glucose-reacting protein 78 (GRP78). The induction of TM inhibits N-acetyl glucosamine-phosphate transmission to UDP-dolichol phosphate, resulting in unfolded proteins. GRP78 then activates the three UPR mediating receptors. The cytosolic portion of ATF6 then enters the nucleus and encodes a homologous C/EBP protein (CHOP) [38]. Overexpression of this CHOP protein leads to autophagy, which inevitably leads to apoptosis (Figure 2).

## 4. Tunicamycin in Cancer Therapy

Despite having some issues, TM’s anticancer properties, which are mediated through UPR induction due to N-linked glycosylation in eukaryotes, are well established [39,40,41]. The following sections briefly and categorically highlight TM’s impact on various cancers. The in vitro and animal studies found that the most targeted tumors for TM are the breast, liver, colon, lungs, and pancreas, depicted in Figure 3.

Breast cancer: Breast cancer (BC) with metastasis, irrespective of the status of receptors (ER, PR, HER-2), are a global threat to female health, as treatment options are still limited and weakly effective. Systemic drug therapy is still the first-line option for metastatic breast cancer treatment with surgery and radiation therapy. These include hormone therapies/endocrine therapies, chemotherapy, targeted therapy, and immunotherapy. Hormone-sensitive patients are treated with drugs that block estrogen production in the body with chemotherapies. These include aromatase inhibitors (anastrozole, letrozole, or exemestane), selective estrogen receptor modulators (SERMs), and other antiestrogen drugs. Women with HER2-positive breast cancer receive an anti-HER2 antibody (trastuzumab) with one more chemotherapy drug, depending on the status of the disease. Recently, HER2-positive patients have been given pertuzumab (anti-HER2 agent) with trastuzumab. Treatment options for triple-negative breast cancer are limited to radiation and chemotherapy. Immunotherapy holds promise for TNBC therapy. Currently, multiple clinical trials are being completed or ongoing to explore the utilization of a combination of immunotherapy with existing therapy for TNBC [42]. However, the current systemic drug therapies most likely cannot remove all cancer cells. Thus, new drugs and therapeutic options are urgently needed.

The therapeutic implication of TM in breast cancer: With several preclinical studies, TM could be a potential drug for treating BC with metastasis, as preclinical studies showed that TM blocks BC growth and metastasis through the regulation of the Akt/NF-kB-signaling pathway [43] and enhances trastuzumab antitumor activity on BC [44], suggesting a promising approach of combination therapy of TM and trastuzumab to improve trastuzumab’s clinical efficiency [44]. The role of Wnt/β-captain in TM-induced ER-stressed mediated apoptosis in triple-negative breast cancer (TNBC) cells has been documented, providing additional TM mechanisms [45].

Cancer stem cells (CSCs) play a significant role in tumor growth and drug resistance [46]. They are characterized to have the cell surface phenotypes CD44+/CD24− [46]. About 1–4% of BC cells carry CD44+/CD24−, which is considered highly aggressive [46]. TM-induced ER stress reduces the growth and invasion of CD44+/CD24− cells, indicating TM therapy is an exciting approach to target breast cancer stem cells [47].

Colon cancer: Colon cancer, also called colorectal cancer (CRC), is one of the most common digestive tract tumors and the leading cause of cancer-related death in the elderly population in the United States and globally [48]. However, early screening can find non-cancerous abnormal growths in the colon, called polyps, which can be removed (polypectomy) and prevent cancerous growth in the colon. In the advanced stage, chemotherapy is the only option before or after radiation. Targeted and immunotherapy is also considered for advanced colon cancer.

Targeting colon cancer by TM: The studies suggest that TM could be an anti-CRC drug, as TM treatment efficiently inhibits CRC cell growth and aggressive behavior by downregulating vimentin, FIB, Ecacollagen type I, Slug expression levels, and survival of tumor-bearing mice [49]. Further, TM has been found to enhance TRAIL-induced apoptosis in CRC cells by JNK-CHOP-mediated DR5 upregulation and inhibiting the EGFR pathway [50]. TM reprogrammed cell–cell adhesion in undifferentiated CRC cells by inducing E-cadherin [51]. Interestingly, separate indirect studies suggested that TM is a promoting factor of CRC [52,53].

Lung cancer: Lung cancer is the most common cancer-related death in the human population and is highly correlated with cigarette smoking [54]. Non-small cell lung cancer (NSCLC) is the primary subtype of lung cancer and accounts for about 80% of the new cases of lung cancer [54,55]. The therapeutic options and efficacies of NSCLC are limited due to drug resistance. Thus, the 5-year survival rate of NSCLC remains at 15% [54,55].

Targeting lung cancer by TM: In lung cancer, the therapeutic roles of TM are well documented. Cisplatin (Cis) has long been a cornerstone therapeutic agent for NSCLC. However, toxicity and resistance to Cis in NSCLC is a common therapeutic obstacle [56]. Multiple pathways mediate Cis resistance in lung cancer; one is the glycosylated form of long pentraxin-3 (PTX3) overexpression in lung cancer cells [57]. PTX3 is an evolutionarily conserved glycoprotein and a member of the Pentraxins superfamily [58]. PTX3 is vital in innate immunity, inflammation, cancer, and tissue repair and remodeling. Depending on the cellular context, PTX3 has protective and pro-tumorigenic effects on various cancers [58]. However, a study showed that TM treatment sensitizes the Cis effect on lung cancer cells, mediated via the deglycosylation of PTX3 through the AKT/NF-κB signaling pathway [55].

Single-cell RNA-sequencing analyses showed that endoplasmic reticular oxidoreductase-1a (ERO1A) correlates with immunosuppression and dysfunction of CD8+ T cells, along with anti-PD-1 treatment and drug resistance. In human lung cancer, high ERO1A expression is associated with a recurrence of the disease following neoadjuvant immunotherapy. Thus, ERO1A suppression in the mouse models imbalances IRE1a and PERK signaling activities, resulting in lethal UPR in tumor cells enduring ER stress. Consequently, anti-tumor immunity via immunogenic cell death is enhanced [59]. Together, studies suggest that TM anti-immunity in a lung cancer mouse model can be enhanced with combination therapy that could disrupt ERO1A [59].

Pancreatic ductal adenocarcinoma (PDAC): Among pancreatic cancers, PDAC is the deadliest cancer in the pancreas and constitutes 90% of pancreatic cancer. PDAC represents the 7th most common cause of cancer-related death globally and has an overall 5-year survival rate not exceeding 9% [60,61]. It is anticipated that PDAC could be the second most common cause of cancer-related death in the USA by 2040 [60,62]. Based on genomic alterations, PDAC can be classified into four groups. These include stable genomes (<50% structural variants per genome), scattered genomes (50–200 structural variants per genome), locally rear-ranged genomes (>200 structural variants clustered on less than three chromosomes), and unstable genomes (>200 structural variants distributed in the genome) [63,64]. This genome classification provides precision oncology knowledge and has clinical potential for PDAC patients [63].

Despite having a limited effect or no outcome of the therapy due to quick development resistance to drugs, the first-line treatments of PDAC are still FOLFIRINOX (a combination of 5FU, leucovirin, oxaliplatin, and irinotecan) and gemcitabine, alone or in combination with nab-paclitaxel [60,65]. The therapeutic limitations of PDAC show the urgency of finding new therapeutic options [60,65]. Mutant K-RAS in the ductal epithelial cells is the driving force of the initiation and progression of PDAC [63,66,67,68,69,70], which can be promoted by multiple factors, including CCN-family proteins [71,72]. Several preclinical studies are actively involved in discovering small molecules to target K-RAS mutants [73].

Mutant K-RAS plays a critical role in the immunosuppression of PDAC microenvironment [74,75,76]. Thus, immunologically, mutant K-RAS has an equally important target [77,78,79]. Despite having mixed and descriptive information on T-helper 2 (Th2) cells and associated cytokines in cancer, recent studies have found that Th2 cells may be crucial regulators of colon and pancreas cancer progression by affecting immune cell composition and type II immune responses. In murine allograft models of colon and pancreatic cancers, the adoptive transfer of Th2 cells into tumor-bearing mice significantly reduces tumor growth by inducing cytotoxicity via eosinophils and macrophages’ innate immune response. The studies also showed that IL-5 protects against tumor growth by recruiting and activating eosinophils [80]. Although glycosylation plays a vital role in regulating the above proteins, the impact of TM is underdetermined.

The overexpression of eukaryotic translational initiation factors in various cancers, including in PDAC, is a common alteration. The studies showed that one of the factors, eIF4E, is overexpressed in poorly differentiated and metastatic PDAC [60,81]. eIF4E phosphorylation (p-eIF4E) at serine209 by MNK1/2 serine/threonine kinases promotes its transformation activity [82,83]. Thus, upon mutant K-RAS activation in PDAC, eIF4E activated and promoted tumor invasion and metastasis.

Targeting PDAC by TM: The role of TM in PDAC is underdetermined compared to other cancers. It is reported that, inherently, PDAC has a high level of ER stress, and that can be promoted by chemotherapy [84]. Further, the studies demonstrated that the ER stress-UPR-lysosomal pathway could be targeted by TM or STF083010 (ER-stress modulator), improving chemotherapy treatment. Furthermore, one study showed that TM treatment can inactivate eIF4E, a downstream signaling molecule of mutant K-RAS in PDAC cells [85]. Based on these studies, we can deduce that TM treatment can be an ideal therapeutic approach for PDAC.

## 5. Tunicamycin in Immunotherapy

Tumor initiation and progression are linked with altered or deluded immunity [86] or the reprogramming of immune cells within the tumor microenvironment [87,88]. The communication between tumor cells and the immune system is dynamic, reminiscent of a balance of immunity factors [86,87]. Thus, immunooncology is now at the forefront of basic, preclinical, and clinical research and has recently been harnessed as one of the pillars of cancer therapy [86,89,90,91,92,93]. The central principle of cancer immunotherapy is eliminating cancer cells by host cytotoxic immune CD8^+^ T cells [86,91,92,93,94]. However, this does not always happen in the tumor ecosystem, as tumor cells manipulate the system, cause tumor-specific CD8+ T cell dysfunction via antigen-derive differentiation program [95], or activate various immune checkpoint suppressive mechanisms. These include regulatory T (Treg) cells and the overexpression of program death-1 (PD-1) receptor and its ligand PD-L1, CTLA-4, and others [90,91,96,97], which are briefly depicted in Figure 4.

In many different malignancies, immune-targeted therapy primarily focuses on the T-cell base [86], as the T-cells have the potency to suppress tumor growth clinically [86,94,98,99]. Despite the clinical successes of immune checkpoint inhibitors, the mechanisms underlying inhibitors’ upregulation in cancer cells and infiltrating T cells have only begun to be understood. The emerging findings showed that indoleamine 2,3-dioxygenase 1 (IDO1) is overproduced in highly tumorigenic self-renewing repopulating cancer cells [100]. IDO1 catalyzes the committing and rate-limiting step of the kynurenine (KYN) metabolic pathway, leading to abundant KYN release [100,101]. KYN is then absorbed in CD8^+^ T cells, activating the aryl hydrocarbon receptor (AhR) that causes PD-1 overexpression [100]. This sequential event adopts a tumor immunological microenvironment that is defective in recognizing and killing cancer cells [101].

The PD-L1 regulation in cancer cells is mediated by multiple pathways depending on the cancer type. These pathways are genomic alterations, epigenetic modifications, transcriptional and post-transcriptional regulation, post-translation modifications, and exosomal transport [102,103,104].

In the tumor microenvironment, the PD-1 receptor is highly N-glycosylated in T cells, thereby maintaining PD-1 stability and interaction with the PD-L1 ligand expressed in tumor cells and dampening the activity of T-cell receptor (TCR)/CD28) signals [105]. The studies showed several N-glycosylated inhibitors, including TM treatment deglycosylated PD-1 [105,106] (Figure 4). Thus, TM could be considered an ideal therapy to target PD-1 alone or in combination with other PD-1 inhibitors.

Like PD-1, PD-L1 is also highly N-glycosylated, maintaining its stability and binding affinity with PD-1 in various cancers and eradicating triple-negative breast cancer (TNBC) by targeting glycosylated PD-L1 [107]. Thus, studies suggest that targeting PD-L1 glycosylation could be an ideal therapeutic option by an appropriate combination of cancer immunotherapies [108,109,110,111].

Interleukins (ILs) are secreted cytokines that play vital roles in the biology of cancers and immune functions [112]. Although some ILs (i.e., IL-6, IL-10, and IL-17) boost tumor growth, some ILs may promote the immune system’s antitumor response. These include IL-2, IL-12, and IL-5 [112,113]. Previous studies have shown that glycosylation plays a role in IL-12 biogenesis, but the loss of glycosylation has little or no effect on IL-12 and IL-23 secretion, heterodimerization, and biological activities [114]. IL-5 is produced by both hematopoietic and non-hematopoietic cells [115]. T-cell-derived IL-5 controls eosinophil production and activation via JAK-STAT and RAS/Raf-ERK signaling pathways [115]. Recent studies have shown that IL-5-producing CD4-positive T cells and eosinophils collectively enhance response to the immune checkpoint blockade (ICB) in breast cancer. Further, it has also been reported that ICB-increased IL-5 production by CD4 T-cells elevated eosinophil production from bone marrow and infiltration that eventually enhanced ICB efficacy [113,116]. T-cell-derived IL-5 exhibits heterogeneous glycosylation with four N-glycosylation sites in the extracellular regions, and removal of N- or O-linked glycosylation on IL-5 enhances the potency of the cytokine [117], without having an effect on lymphocyte proliferation in a lymphocyte proliferation assay following the treatment of the glycosylation inhibitor TM [115,118]. Previously, however, it has been shown that, in humans, IL-5 acts only on eosinophils and basophils for their growth, maturation, and activation, as these cells express IL-5 receptors [119,120,121]. Thus, the results of these studies suggest that the inhibition of glycosylation in IL-5 by TM could be effective only on eosinophil production and maturation, and their activation. However, further studies are warranted.

In pancreatic cancer progression, IL-5/IL-5Rα roles are possibly a double-edged sword, with both pro- and antitumor activities recently discovered [80,122,123,124]. However, the mechanistic roles of IL-5 are still unclear. Studies have shown that TM can inhibit endoplasmic-reticulum (ER)-stress-mediated autophagy and promote the effectiveness of chemotherapies on pancreatic cancer cells [84]. Further, studies have shown that ER-stressed inhibitors reduce IL-5 production in Th-2 cells (naïve CD4+ T cells) [125], which inhibit colon and pancreatic cancer growth by promoting antitumorigenic responses from macrophages and eosinophils [80]. However, the effect of ER-stressed inhibitors, including TM, on IL-5 in pancreatic cancer cells and localized eosinophils is still uncertain.

In conclusion, TM shows promise in drug-induced immunotherapy when combined with other PD-1 and PD-L1 inhibitors, such as monoclonal antibodies or various chemotherapeutics (Figure 4).

## 6. The Pitfalls of the Use of Tunicamycin

Basic and preclinical studies showed that TM could be a promising cancer drug, as it kills various primary cancer cells and effectively enhances the efficacy of chemotherapy and immunotherapy. However, in reality, it could not ensue because of the TM-induced toxicity in various organs [126]. The direct administration of TM in murine models of leukemia resulted in potent liver toxicity. Further, TM treatment caused kidney and liver damage in rats. TM can cause severe and often fatal neurological toxicity in animals, which is known as annual ryegrass toxicity [127]. The studies have also shown that TM causes apoptosis in peripheral neurons [128]. This makes TM a liability in its free form, and its entry into the clinic is thus forbidden or restricted [126]. However, a study demonstrated that the toxicity of TM can be reduced by modifying the chemical structures [129], indicating a future promise of the use of TM as a therapy. Further studies are warranted.

## 7. Can Nanoencapsulation Overcome the Pitfall?

As described in preceding sections, a notable characteristic of TM lies in its reliable capacity to inhibit N-linked glycosylation, a process that has demonstrated significant antitumor efficacy [4,19]. However, a prominent pitfall is its lack of cell type and tissue-specific toxic effects, posing a substantial life-threatening risk in TM therapy. Consequently, the advancement of TM research in clinical settings has stagnated, underscoring the urgent need for novel strategies to leverage TM therapeutically in cancer treatment.

Nanoparticles have emerged as versatile tools, finding applications across diverse domains ranging from cosmetics [130,131] to drug delivery [132,133], and have even been used to clean the environment [130,131]. These minuscule entities have revolutionized modern cancer therapy, enhancing its efficacy and specificity manifold, notably in radiotherapy [134] and chemotherapy [135].

A diverse array of nanoparticles serves to fine-tune the absorption, distribution, metabolism, and elimination (ADME) profile of small molecule drugs, thereby optimizing their pharmacokinetic and dynamic behavior to benefit the patient [136]. These nanoparticles typically consist of drug–polymer conjugates, drug–antibody conjugates, drug-encapsulating micelles, or polymersomes. Among these systems, conjugates are typically synthesized through covalent bonding, while micelles and polymersomes encapsulate drugs via non-covalent assembly [137]. The selection of nanoscale delivery systems is contingent upon the physicochemical properties of the drug in question, ensuring optimal compatibility and performance.

Stimuli-responsive nanoparticles, remarkably those capable of reacting to specific environmental cues, are another significant front of drug delivery [138]. These systems can only be tailored to release their drug payload in response to cellular or biochemical signals relevant to a particular disease [139]. It is worth noting that nanoscale systems not only hold the potential to revolutionize drug delivery but also offer promising prospects in diagnostics and biomarker detection, representing emerging and crucial outcomes of nanotechnology. In the context of TM, nanoparticle technology can potentially enhance the drug’s volume of distribution (V_D_). By facilitating localized accumulation, TM delivered via nanoparticles can markedly reduce TM-associated toxicity, similar to earlier work in the field [140]. The kinetics of TM release from within the nanoparticles govern the plasma accumulation of the drug. In simple encapsulated systems, drug release from nanoparticles typically follows Fick’s law for diffusion. Nanosystems connect the drug via covalent linkage. The breakdown of the chemical backbone dictates the rate and extent of drug release. Therefore, when carefully designed, nanoencapsulation and chemical conjugation can effectively control the cytotoxic side effects and drug delivery [141]. For example, free-form irinotecan produced severe side effects due to a very high plasma concentration, while encapsulated irinotecan treatment showed less or no burst effect and toxicity and exhibited excellent antitumor efficacy. It was thus strongly recommended as a rectal pharmaceutical product alternative to commercial intravenous injection in the treatment of rectum and colon cancer [141]. Cancerous tissue has many pathophysiological anomalies compared to healthy tissues. These include leaky vasculature and poor lymphatic drainage. The nanoparticles benefit from their small size and accumulation [132,142,143]. As an effective and cytotoxic drug, TM is the perfect candidate for nanoencapsulation for therapeutic implications. Thus, further studies are warranted.

Gold nanoparticles (GNPs) have been used to encapsulate TM and target capillary endothelial cells [144]. Cells were treated by free and encapsulated TM to determine the difference in the efficacy of cell cycle arrest. Nanoencapsulated TM was found to have 50% more efficacy than free-form TM. The effect of TM on N-linked glycosylation was also studied. They found that dolichol phosphate mannose synthase (DPMS), a protein instrumental in protein folding, was decreased by 33% when treated with free TM, while cells exposed to encapsulated TM showed a significantly high (about 66%) impact. Thus, this study proposed the nanoencapsulation of TM as possibly being a real asset in an effective TM treatment. Given the importance of the nanoencapsulation of TM, tumor-specific hypoxic/pH-responsive nanoparticles could be an ideal approach to targeting primary tumors and the surrounding tumor ecosystem/microenvironment [145,146].

Further, the chemical conjugation of TM to an antibody or a macromolecule like PEG can result in the formation of antibody–drug conjugates (ADCs) or polymer–drug conjugates (PDCs), respectively. Both variants can augment the cellular accumulation of TM inside cancer cells. Collectively, nanotechnology can also provide exciting prospects that may improve TM’s implications with cytotoxicity and efficiency [146,147,148,149].

## 8. Outlook

TM shows promise if it is to be proven applicable in targeted therapy; however, more research and trials in mice models are needed before it can be approved for the clinic. Because of TM’s toxicity and off-target effect, many precautions, such as kidney and liver toxicity, bioavailability, and target-specific drug delivery, need to be taken regarding nanoparticle delivery. These challenges are not expected to be resolved promptly.

## Figures and Tables

**Figure 1 cells-13-00395-f001:**
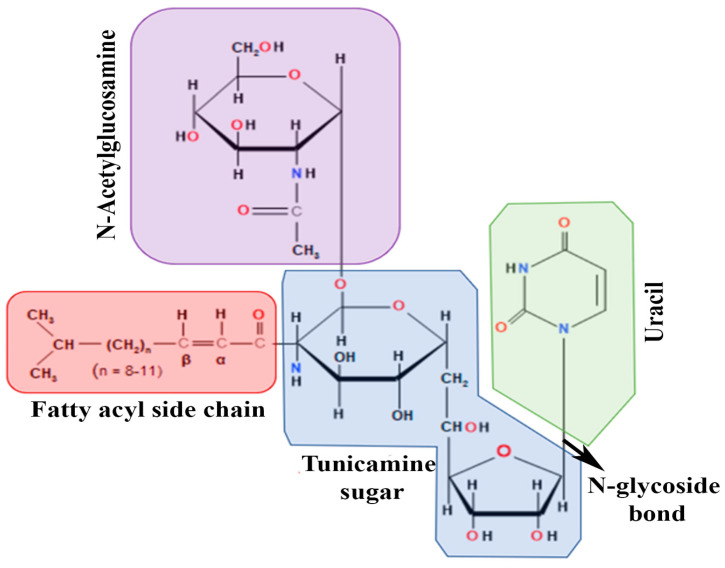
Chemical structure of Tunicamycin. The molecule consists of various structural fragments crucial in its anti-cancer properties. The first fragment of interest is the aminosugar moiety, which consists of a hexose sugar called N,N′-diacetyl chitobiose. This fragment is responsible for the initial recognition and binding of tunicamycin to its target enzyme, UDP-N-acetylglucosamine-dolichyl-phosphate N-acetylglucosaminephosphotransferase (GPT). The interaction between the aminosugar and the enzyme is crucial for inhibiting the biosynthesis of N-linked glycoproteins, a process essential for cancer cell growth and metastasis. Another vital fragment is the nucleoside moiety, which consists of a thymidine derivative known as 4-(2-amino-2-deoxy-α-D-glucopyranosyl) thymine. It stabilizes the interaction between tunicamycin and GPT, thereby enhancing the drug’s potency. The third fragment of interest is the fatty acid chain, which consists of a 17-carbon isoprenoid structure, sometimes called decaprenyl phosphate. It is involved in anchoring the drug to the endoplasmic reticulum, the site of glycoprotein synthesis, facilitating its localization to the target enzyme. The fatty acid chain for various derivatives of tunicamycin, composed of 14–17 carbon chain lengths, contributes to its overall hydrophobicity, which affects its pharmacokinetic properties. Finally, the unsaturated uridine ring system binds to the target enzyme and contributes to its inhibitory potency.

**Figure 2 cells-13-00395-f002:**
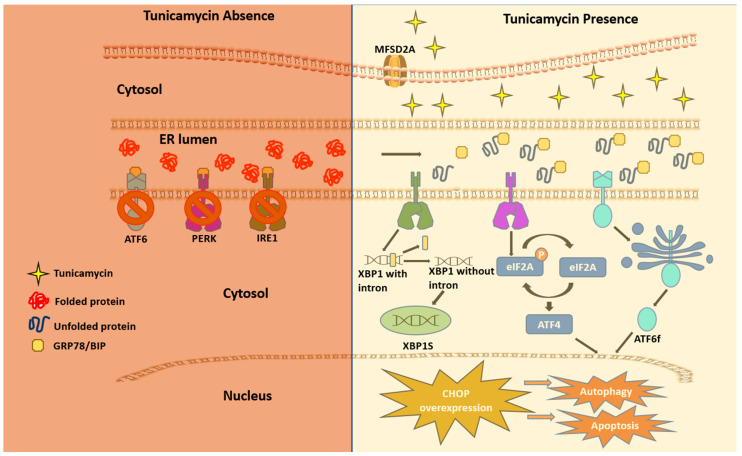
The unfolded protein response (UPR) via TM. This figure presents a cell in two settings: normal homeostasis (depicted by a crimson-shaded region) and TM presence (depicted by a beige-shaded region). In the crimson-shaded section, GRP78/BiP binds to IRE1, PERK, and ATF6, the three sensors of the UPR, and as a result, downstream signals are blocked. The beige section presents how Tu stimulates the UPR. The MFSD2A gene encodes a plasma membrane transporter to perform the cellular uptake of TM, which disrupts the glycosylation process by inhibiting GPT and assists in the first step of glycosylation, resulting in the blockage of dolichol phosphate. This allows unfolded proteins to accumulate in the ER lumen, and GRP78/BiP chaperone detaches itself from the three UPR sensors to bind to newly synthesized unfolded proteins. IRE1 goes through a cytoplasmic splicing of its mRNA, resulting in an XBP1 polypeptide. XBP1 then assists in ER stress-induced autophagy. The phosphorylation of eIF2a inhibits translation initiation, a part of the translation process. Under ER stress, ATF6 (activating transcription factor 6) is transported to the Golgi body, where cleavage is performed on ATF6, becoming a functional transcription factor (ATF6f). These three UPR stimulants, XBP1S, phosphorylated eIF2a, and ATF6f, enter the nucleus. A nucleic response ends in CHOP overexpression, leading to either autophagy or apoptosis.

**Figure 3 cells-13-00395-f003:**
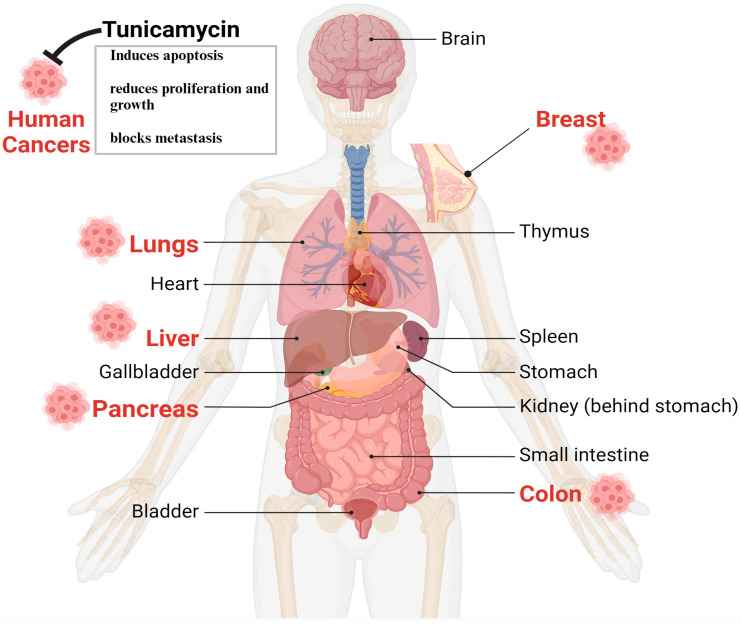
The therapeutic impact of TM on human various cancer cells and animal models. Based on in vitro and preclinical animal studies, TM’s antitumorigenic effects are found in the lungs, liver, breast, pancreas, and colons. Thus, TM can be an ideal drug to treat human cancers in combination with other antibodies or chemotherapies.

**Figure 4 cells-13-00395-f004:**
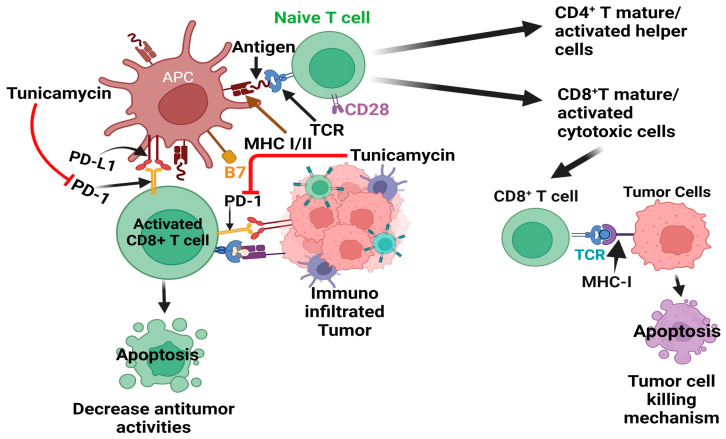
Effect of tunicamycin on tumor cell killing by blocking PD-1-PD-L1 axis inhibition. The antigen-producing cells (APC) promote the maturation/activation of naïve T cells. The active CD8+ T cells, via their receptors (TCR), bind with tumor cell-producing antigens with MHC I and activate tumor cell-killing mechanisms. However, aggressive tumor cells produced program cell death ligand 1 (PD-L1), which binds with its receptor PD-1 in CD8+ T cells and deactivates T cells, which decreases the antitumor activities of T cells.

## Data Availability

Not applicable.

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
