# Peer review of "Benefits and Pitfalls of a Glycosylation Inhibitor Tunicamycin in the Therapeutic Implication of Cancers"

_cells, 2024, doi:10.3390/cells13050395_

Round 1

Reviewer 1 Report

Comments and Suggestions for Authors

The authors summarized the benefits and pitballs of tunicamycin in therapeutic implication of cancers. This review is of great importance, especially for researchers in cancer research and glycobiology. The work is well organized and contains sufficient reference to related and previous work.

There are few minor corrections:

Line 50, change the capital G of Glycosylation.

Line 51, are these enzymes glycotransferases and/or glycosidase? Please specify

Please check the use of abbreviation TM.  

In line the content between line 351 to 362.

Author Response

There are a few minor corrections:

Line 50, change the capital G of Glycosylation.

RESPONSE:  Done

Line 51, are these enzymes glycosyltransferases and/or glycosidase?  Please specify

RESPONSE: Multiple enzymes can be used as biomarkers.  Glycosyltransferases and glycosidases are among them.   They are responsible for the assembly, processing, and turnover of glycans.  In addition, several transferases modify glycans by adding acetyl, methyl, phosphate, sulfate, and other groups.

Please check the use of abbreviation TM.  

In line the content between line 351 to 362.

RESPONSE:  Done

Reviewer 2 Report

Comments and Suggestions for Authors

The authors present a review regarding Tunicamycin for cancer therapy. Firstly, the molecule is described in detail and its mode of action is described. Then different (potential) applications in various types of cancer are discussed.

A comprehensive overview of this topic is an interesting approach, however, the presentation has to be improved drastically. The collection of references is well done, but the extraction of the content is weak. Sometimes methodical details are given (example chapter 2), sometimes just the title of the paper is copied with no further interpretation. Especially in chapter 4 it is very often not clear, if Tunicamycin is already used or an untested option.

Furthermore, the language and several typos hamper the reading.

In detail:

Major:

·         Rephrase the abstract (line 28-34). It is not clear what is presented here and what is presented somewhere else.

·         Line 41: just asparagines

·         Rephrase and restructure whole chapter 4. I would suggest for each type of cancer: first a description of the respective cancer, then the current therapies, and then everything around TM. Make clear what is already in use and what is just under consideration.  

·         Due to the fact that TM affects all types of cell, chapters 6 & 7 should give more detailed information how to overcome this challenge.

Minor:

·         Line 22: … , specialized macromolecules instrumental in N-linked glycosylation – what does this mean?

·         References [1,2] cover just the starting point of O-glycosylation; chose better references to cover whole N- and O-glycosylation.

·         Line 69: the resulting proteins are not necessarily unfolded.

·         Fig 1 Remove “Tunicamycine” from the figure. It is the name for the whole molecule and not just a part of it.

·         Legend Fig 1 Line 104: Correct N-glycosylation is an essential process for all cells not only for cancer cells.

·         Line 113: Language!

·         Fig 2 Give an explanation in the legend

·         Page 5 Line 176-188 TM is the abbreviation for tunicamycin.

·         Legend Fig 3: Headline twice.

Comments on the Quality of English Language

Language has to be improved

Author Response

Major:

  • Rephrase the abstract (line 28-34). It is not clear what is presented here and what is presented somewhere else.

RESPONSE:  We modified the statement and clearly described the review's objectives.

 Line 41: just asparagines

RESPONSE:  Modified it.

Rephrase and restructure whole chapter 4. I would suggest for each type of cancer: first a description of the respective cancer, then the current therapies, and then everything around TM. Make clear what is already in use and what is just under consideration.

RESPONSE:  We modified Chapter 4 as per the reviewer's suggestions.    

Due to the fact that TM affects all types of cell, chapters 6 & 7 should give more detailed information how to overcome this challenge.

RESPONSE:  We modified these two chapters carefully and explained them elaborately.

Minor:

Line 22: … , specialized macromolecules instrumental in N-linked glycosylation – what does this mean?

RESPONSE:  We remove the sentence from the revised manuscript. 

References [1,2] cover just the starting point of O-glycosylation; chose better references to cover whole N- and O-glycosylation.

RESPONSE:  Added two partinent references. 

Line 69: the resulting proteins are not necessarily unfolded.

RESPONSE:  Modified the sentence. 

Fig 1 Remove "Tunicamycine" from the figure. It is the name for the whole molecule and not just a part of it.

  • Legend Fig 1 Line 104: Correct N-glycosylation is an essential process for all cells not only for cancer cells.

Line 113: Language!

RESPONSE:  The question is not clear to us.

Fig 2 Give an explanation in the legend

RESPONSE:  Legend added.       

Page 5 Line 176-188 TM is the abbreviation for tunicamycin.

RESPONSE:  Done.

 Legend Fig 3: Headline twice.

RESPONSE:  Deleted. 

Round 2

Reviewer 2 Report

Comments and Suggestions for Authors

The revised version of this paper has slightly improved. Two necessary publications were added and the figure legends are now much more informative.

However, the publication still does not fulfill the expectations raised in the abstract. For example, it says "Finally, we discuss the potential use of nano-based drug delivery systems…". In fact, the topic is not discussed but only dealt with in one sentence. Chapters 6 + 7 are still very thin in terms of content.

A few additions have been made to chapter 4, but overall it is still very confusing.

The revision was obviously carried out in great haste. Although the commentary states that the correction was made from TN to TM, this was not done (now line 212-217)

I think, the authors need more time to improve the publication significantly.

Comments on the Quality of English Language

Improved compared to the previous version

Author Response

Based on the reviewer's comments and suggestions, we have modified the manuscript extensively.  We hope that the reviewers will accept the modification we made in the revised manuscript.  The revised manuscript (word document) is attached.  If you have any questions or concerns, please me know.

Thank you again for giving us an opportunity to revised the manuscript.

Sincerely

Sushanta Banerjee
